# Type 2 Diabetes Mellitus- and Complication-Related Risk of Nontuberculous Mycobacterial Disease in a South Korean Cohort

Da Som Jeon,[a] Seonok Kim,[b] Mi Ae Kim,[c] Yong Pil Chong,[d] Tae Sun Shim,[e] Chang Hee Jung,[f] Ye-Jee Kim,[b] Kyung-Wook Jo[e]

aDivision of Pulmonology and Critical Care Medicine, Department of Internal Medicine, Nowon Eulji Medical Center, University of Eulji, Seoul, Republic of Korea

bDepartment of Clinical Epidemiology and Biostatistics, Asan Medical Center, University of Ulsan College of Medicine, Seoul, Republic of Korea

cDepartment of Internal Medicine, Dongsan Medical Center, Keimyung University School of Medicine, Daegu, Republic of Korea

dDepartment of Infectious Diseases, Asan Medical Center, University of Ulsan College of Medicine, Seoul, Republic of Korea

eDivision of Pulmonary and Critical Care Medicine, Department of Internal Medicine, Asan Medical Center, University of Ulsan College of Medicine, Seoul, Republic of Korea

fDivision of Endocrinology and Metabolism, Department of Internal Medicine, Asan Medical Center, University of Ulsan College of Medicine, Seoul, Republic of Korea

Da Som Jeon, Seonok Kim, and Mi Ae Kim contributed equally to the manuscript. Author order was determined by drawing straws.

**ABSTRACT** We aimed to investigate whether type 2 diabetes mellitus (T2DM) and diabetes-related complications constitute significant risk factors for nontuberculous mycobacterial (NTM) disease. Data from the National Health Insurance Service-National Sample Cohort (which represents 2.2% of the total South Korean population) recorded between 2007 and 2019 were extracted to establish the NTM-naive T2DM cohort ($n$ = 191,218) and the 1:1 age- and sex-matched NTM-naive matched cohort ($n$ = 191,218). Intergroup comparisons were performed to determine differences in the NTM disease risk of the two cohorts during the follow-up period. During median follow-up of 9.46 and 9.25 years, the incidence of NTM disease was 43.58/100,000 and 32.98/100,000 person-years in the NTM-naive T2DM and NTM-naive matched cohorts, respectively. Multivariable analysis showed that T2DM alone did not confer a significant risk for incident NTM disease, although T2DM with ≥2 diabetes-related complications significantly increased NTM disease risk (adjusted hazard ratio [95% confidence interval], 1.12 [0.99 to 1.27] and 1.33 [1.03 to 1.17], respectively). In conclusion, the presence of T2DM with ≥2 diabetes-related complications significantly increases the risk for NTM disease.

**IMPORTANCE** We assessed whether patients with T2DM are at higher risk for incident NTM disease through analysis of NTM-naive matched cohorts from the data of a national population-based cohort which represents 2.2% of the total South Korean population. Although T2DM alone is not a statistically significant risk factor for NTM disease, T2DM significantly increases the risk of NTM disease in those with ≥2 diabetes-related complications. This finding suggested that patients with T2DM with a larger number of complications should be considered a high-risk group for NTM disease.

**KEYWORDS** type 2 diabetes mellitus, complication, *Mycobacterium* infections, nontuberculous mycobacteria, nontuberculous

Epidemiological data clearly show a global increase in the incidence and prevalence of nontuberculous mycobacterial (NTM) disease (1–3). In South Korea, several studies have consistently reported a substantially increasing trend of NTM disease in the last 15 years (1, 4). Several factors, such as technical advances in the isolation and identification of NTM from clinical specimens, wider recognition of NTM as a human pathogen, overall population aging, decreased immune function caused by various medical treatments for cancer or

Address correspondence to Ye-Jee Kim, kimyejee@amc.seoul.kr, or Kyung-Wook Jo, siegliede@gmail.com.

The authors declare no conflict of interest.

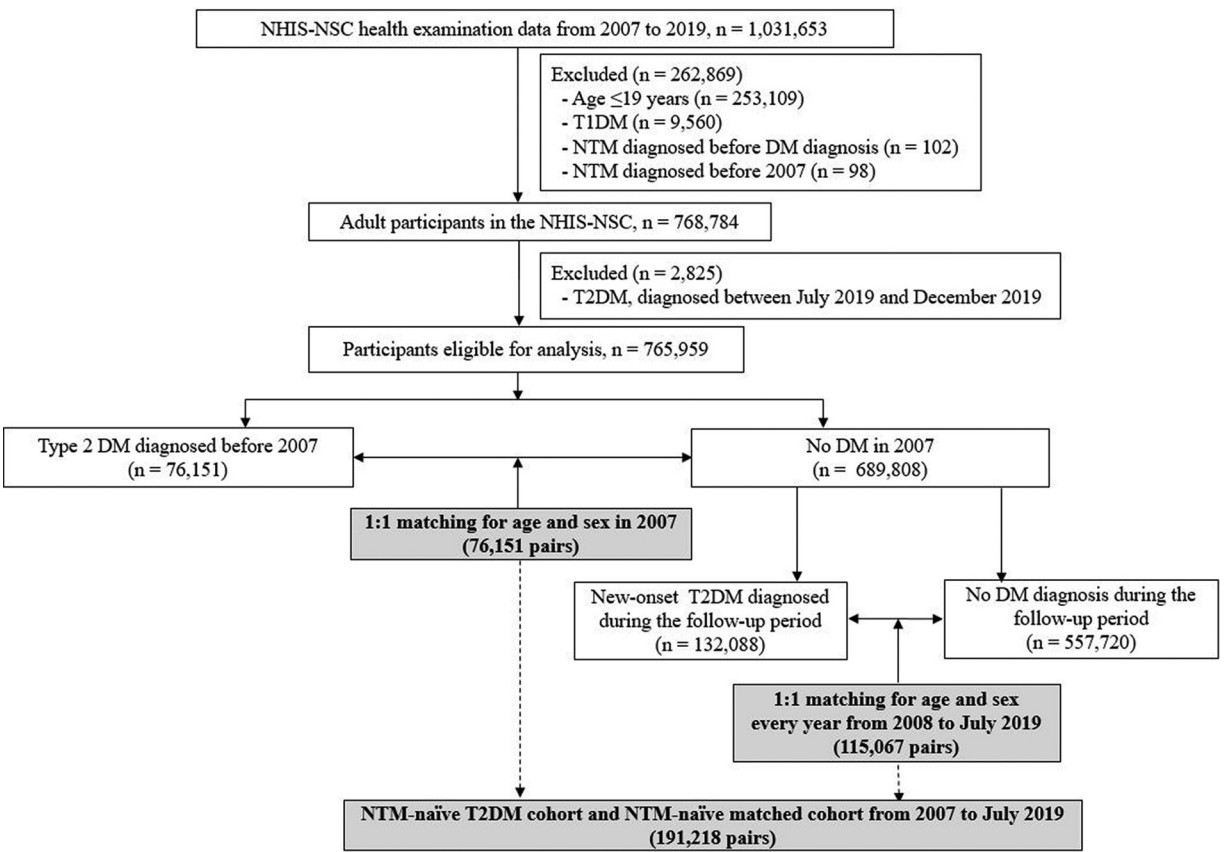

**FIG 1** Flow chart of the study population. Abbreviations: NHIS-NSC, National Health Insurance Service-National Sample Cohort; T1DM, type 1 diabetes mellitus; NTM, nontuberculous mycobacterium; T2DM, type 2 diabetes mellitus.

rheumatologic disease, and frequent exposure to environmental mycobacteria, have been implicated as possible causes of the increase in NTM disease in the past decades (5–8).

Diabetes mellitus (DM) is one of the most common diseases in South Korea, and the average annual prevalence of DM and impaired fasting glucose (IFG) in Korean adults aged ≥20 years is 13.8% and 28%, respectively (9). Oh et al. reported that most South Korean diabetes patients had type 2 DM (T2DM), which accounts for 88.0% of all cases of diabetes (10). DM increases the severity, incidence, and complications of infectious diseases (11), as DM impairs macrophage or T-cell function and cytokine production (12, 13). Tuberculosis is an important infectious disease that is associated with diabetes (14). Compared with those without diabetes, patients with diabetes have a severalfold higher risk for tuberculosis (15, 16). Based on an analysis of health insurance data, Golub et al. reported that South Korean DM patients had a higher risk for tuberculosis than those without DM (17).

Patients with NTM disease and tuberculosis have several similar clinical characteristics, including clinical symptoms, radiographic findings, and treatment regimens (18–20); however, studies of the impact of DM on incident NTM disease are limited (21). Using a nationwide longitudinal cohort in South Korea, we aimed to assess whether T2DM, the major form of DM in Korean adults, is a significant risk factor for NTM disease. Furthermore, we investigated whether the NTM risk differs in the presence of diabetes-related complications.

## RESULTS

**Study subjects and matched cohort.** Through eligibility screening and a 1:1 matching process for age and sex, we established the NTM-naive T2DM cohort (*n* = 191,218) and the NTM-naive matched cohort (*n* = 191,218) (Fig. 1). As shown in Table 1, the two groups were

**TABLE 1** Year of diagnosis of diabetes mellitus and the baseline characteristics of the type 2 diabetes mellitus cohort and matched cohort[a]

| Characteristic | NTM-naive T2DM cohort (n = 191,218) | NTM-naive matched cohort (n = 191,218) | P value |
|---|---|---|---|
| Yr of diagnosis of T2DM | | | |
| Before 2007 | 76,151 (39.82) | | |
| 2007 | 10,969 (5.74) | | |
| 2008 | 9,735 (5.09) | | |
| 2009 | 9,607 (5.02) | | |
| 2010 | 9,309 (4.87) | | |
| 2011 | 9,511 (4.97) | | |
| 2012 | 9,039 (4.73) | | |
| 2013 | 9,431 (4.93) | | |
| 2014 | 9,007 (4.71) | | |
| 2015 | 8,836 (4.62) | | |
| 2016 | 8,943 (4.68) | | |
| 2017 | 8,648 (4.52) | | |
| 2018 | 8,302 (4.34) | | |
| January 2019–June 2019 | 3,730 (1.95) | | |
| Age, yr | | | |
| ≤39 | 18,501 (9.68) | 19,656 (10.28) | |
| 40–49 | 37,900 (19.82) | 40,488 (21.17) | |
| 50–59 | 57,349 (29.99) | 59,624 (31.18) | |
| 60–69 | 44,491 (23.27) | 40,892 (21.39) | |
| ≥70 | 32,977 (17.25) | 30,558 (15.98) | |
| Sex | | | |
| Male | 97,676 (51.08) | 97,676 (51.08) | |
| Female | 93,542 (48.92) | 93,542 (48.92) | |
| Smoking status | | | |
| Current or ever smoker | 78,636 (41.12) | 73,600 (38.49) | <0.001 |
| Comorbidity | | | |
| Charlson comorbidity index | | | |
| 0 | 61,506 (32.17) | 117,617 (61.51) | <0.001 |
| 1 | 53,523 (27.99) | 45,026 (23.55) | |
| ≥2 | 76,189 (39.84) | 28,575 (14.94) | |
| Hypertension | 79,901 (41.79) | 36,925 (19.31) | <0.001 |
| Dyslipidemia | 61,978 (32.41) | 22,880 (11.97) | <0.001 |
| Asthma | 23,062 (12.06) | 14,869 (7.78) | <0.001 |
| Malignancy | 8,098 (4.23) | 4,627 (2.42) | <0.001 |
| Chronic obstructive pulmonary disease | 5,997 (3.14) | 3,328 (1.74) | <0.001 |
| Bronchiectasis | 1,603 (0.84) | 946 (0.49) | <0.001 |
| Chronic kidney disease | 1,750 (0.92) | 306 (0.16) | <0.001 |
| Health insurance type | | | |
| Employee health insurance | 77,000 (40.27) | 76,248 (39.87) | <0.001 |
| Self-employed health insurance | 103,879 (54.32) | 108,316 (56.65) | |
| Medical aid | 10,339 (5.41) | 6,654 (3.48) | |
| Follow-up duration (mo) | 9.25 (4.73–13.00) | 9.46 (4.92–13.00) | <0.001 |

[a]Data are presented as the number (percentage) and median (interquartile range). Abbreviations: NTM, nontuberculous mycobacterium; T2DM, type 2 diabetes mellitus.

well matched in terms of age and sex. In the T2DM cohort, a total of 39.82% of patients were diagnosed with DM before 2007, and the remaining patients were incident cases, diagnosed with DM after 2007 (Table 1). A higher number of patients with current or smoking history were present in the T2DM cohort than in the matched cohort (41.12% versus 38.49%, $P < 0.001$). With regard to comorbidities, the proportions of patients with hypertension, dyslipidemia, bronchial asthma, malignancy, chronic obstructive pulmonary disease, bronchiectasis, and chronic kidney disease were all significantly higher in the T2DM cohorts than in the matched cohort (Table 1). The median follow-up duration of patients in the T2DM cohort and matched cohort was 9.46 (interquartile range [IQR], 4.92 to 13.00) and 9.25 (IQR, 4.73 to 13.00) years, respectively.

**TABLE 2** Treatment modality and diabetes-related complications in the type 2 diabetes mellitus cohort[a]

| Variable | NTM-naive T2DM cohort (*n* = 191,218) |
|---|---|
| Treatment of T2DM | |
| No treatment | 129,725 (67.84) |
| Treatment | 61,493 (32.16) |
| Oral antidiabetes agents | 51,743 (27.06) |
| Insulin | 9,750 (5.10) |
| Classification of oral antidiabetes agents[b] | |
| Metformin | 39,942 (20.89) |
| Sulfonylurea | 37,439 (19.58) |
| DPP4 inhibitor | 7,741 (4.05) |
| $\alpha$-Glucosidase inhibitor | 7,219 (3.78) |
| Thiazolidine | 3,891 (2.03) |
| Meglitinide | 1,363 (0.71) |
| SGLT2 inhibitor | 984 (0.51) |
| | |
| Complication of T2DM | |
| Type of complication | |
| Neuropathy | 49,361 (25.81) |
| Peripheral vascular disease | 42,961 (22.47) |
| Cardiovascular disease | 36,439 (19.06) |
| Cerebrovascular disease | 24,304 (12.71) |
| Nephropathy | 23,268 (12.17) |
| Retinopathy | 16,419 (8.59) |
| No. of complications | |
| ≤1 complication | 138,813 (72.59) |
| ≥2 complications | 52,405 (27.41) |
| Total no. of complications | |
| 0 | 76,422 (39.97) |
| 1 | 62,391 (32.63) |
| 2 | 33,285 (17.41) |
| 3 | 13,805 (7.22) |
| 4 | 4,303 (2.25) |
| 5 | 908 (0.47) |
| 6 | 104 (0.05) |

[a]Data are presented as the number (percentage). Abbreviations: NTM, nontuberculous mycobacterium; T2DM, type 2 diabetes mellitus; DPP, dipeptidyl peptidase; SGLT, sodium-glucose cotransporter.
[b]Numbers are not mutually exclusive.

**Antidiabetes drugs and complications among T2DM patients.** A detailed classification of antidiabetes drug prescriptions and complications among 191,218 patients in the T2DM cohort is shown in Table 2. A total of 32.16% patients were treated with oral antidiabetes agents or insulin. Among the oral antidiabetes agents, the most frequently prescribed drug was metformin (20.89%), followed by sulfonylurea (19.58%) and dipeptidyl peptidase IV inhibitor (4.05%). Additionally, 5.1% of patients were treated with insulin, with or without oral antidiabetes agents.

With regard to diabetes-related complications, 114,796 patients (60.03%) had at least one diabetes-related complication. In addition, a total of 27.41% patients had ≥2 complications. As Table 2 shows, the most frequently noted T2DM complication was neuropathy (25.81%), followed by peripheral vascular disease (22.47%).

**NTM disease risk associated with T2DM.** Among the 191,218 patients in the T2DM cohort, NTM disease developed in 709 persons during the follow-up period, which corresponds to the overall incidence of NTM disease as 43.58/100,000 person-years (PY) (Table 3). Among the matched cohort, the development of NTM disease was noted in 546 patients, indicating an incidence of 32.98/100,000 PY. The univariate hazard ratio (HR) for NTM disease development of T2DM was 1.32 (95% confidence interval [CI], 1.18 to 1.48, $P < 0.001$). Figure 2A shows that the cumulative incidence of NTM disease was significantly higher in the T2DM cohort than that in the matched cohort across all years of tracking ($P < 0.001$).

**TABLE 3** Results of the univariate and multivariable analyses of nontuberculous mycobacterial disease by type 2 diabetes mellitus and the number of diabetes-related complications[a]

| Setting | Total no. | Matched pair | NTM-naive T2DM cohort | | | NTM-naive matched cohort | | | Crude HR (95% CI) | P value | Adjusted HR (95% CI)[b] | P value |
|---|---|---|---|---|---|---|---|---|---|---|---|---|
| | | | NTM disease | Person-yr | Incidence (95% CI) | NTM disease | Person-yr | Incidence (95% CI) | | | | |
| T2DM vs no T2DM | 382,436 | 191,218 | 709 | 1,626,902.08 | 43.58 (40.43–46.91) | 546 | 1,655,672.76 | 32.98 (30.27–35.86) | 1.32 (1.18–1.48) | <0.001 | 1.12 (0.99–1.27) | 0.074 |
| T2DM with ≤1 complication vs no T2DM | 277,626 | 133,813 | 474 | 1,215,019.99 | 39.01 (35.58–42.69) | 396 | 1,231,953.23 | 32.14 (29.06–35.47) | 1.22 (1.06–1.39) | 0.004 | 1.05 (0.91–1.22) | 0.497 |
| T2DM with ≥2 complications vs no T2DM | 104,810 | 52,405 | 235 | 411,882.09 | 57.06 (49.99–64.84) | 150 | 423,719.53 | 35.40 (29.96–41.54) | 1.61 (1.32–1.98) | <0.001 | 1.33 (1.03–1.70) | 0.027 |

[a]Abbreviations: NTM, nontuberculous mycobacterium; T2DM, type 2 diabetes mellitus; HR, hazard ratio; CI, confidence interval.
[b]Adjusted covariates included age, sex, smoking, Charlson comorbidity index, bronchiectasis, chronic obstructive pulmonary disease, chronic kidney disease, asthma, malignancy, and dyslipidemia.

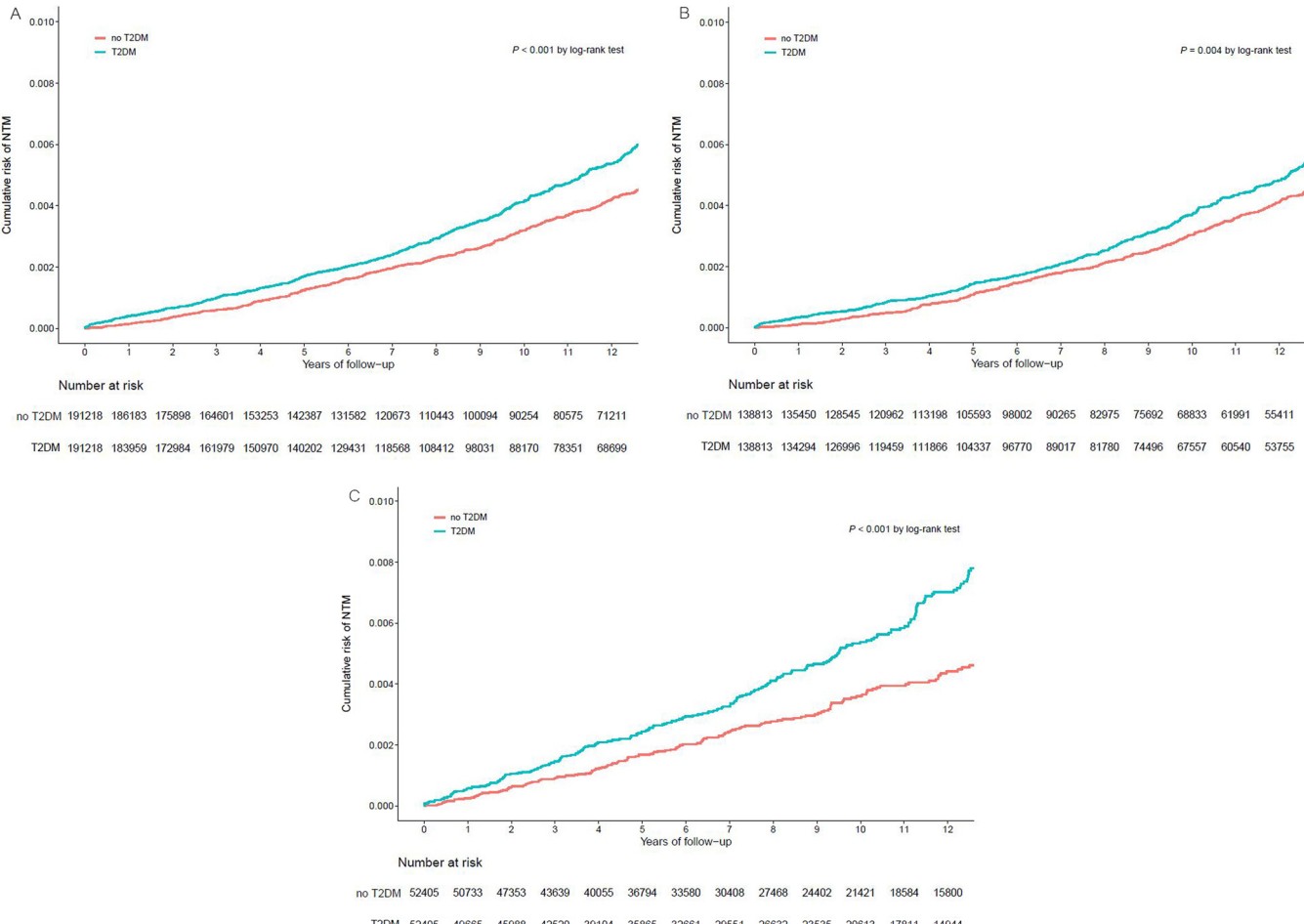

**FIG 2** Kaplan-Meier plot of the cumulative risk of nontuberculous mycobacterial disease in the entire type 2 diabetes mellitus (T2DM) cohort and the matched cohort (A), T2DM with ≤1 complication and the matched cohort (B), and T2DM with ≥2 complications and the matched cohort (C).

Several covariates, including age, sex, smoking status, bronchiectasis, chronic obstructive pulmonary disease, bronchial asthma, dyslipidemia, malignancy, chronic kidney disease, and Charlson comorbidity index (CCI), were related to NTM disease in univariate analyses (see Table S1 in the supplemental material). After the adjustment of these confounding factors, the statistical significance of the impact of T2DM on NTM disease risk did not persist; the adjusted HR of T2DM for NTM disease risk was 1.12 (95% CI, 0.99 to 1.27, $P = 0.074$).

**Incident NTM disease according to the number of diabetes-related complications.** As Table 3 shows, the univariate analysis showed that T2DM was a significant risk factor for incident NTM disease regardless of the number of complications; the HR was 1.22 (95% CI, 1.06 to 1.39, $P = 0.004$) in patients with ≤1 complication and 1.61 (95% CI, 1.32 to 1.98, $P < 0.001$) in those with ≥2 complications. The cumulative incidence of NTM disease according to the number of complications based on the Kaplan-Meier method is shown in Fig. 2B and C, respectively.

However, multivariable analysis revealed that statistical significance was noted only in the case of a higher number of diabetes-related complications; the adjusted HR was 1.05 (95% CI, 0.91 to 1.22, $P = 0.497$) in patients with either no or 1 diabetic complication, whereas the HR significantly increased to 1.33 (95% CI, 1.03 to 1.170, $P = 0.027$) in those with ≥2 complications of DM.

## DISCUSSION

The incidence and prevalence of NTM disease have recently increased worldwide (1–3), and DM is one of the most common chronic diseases whose prevalence has increased

in the last several decades (22). However, few studies have investigated the relationship between DM and the incidence of NTM disease so far (21). In the present study, through analysis of NTM-naive matched cohorts from the data of a national population-based cohort which represents 2.2% of the total South Korean population, we assessed whether patients with T2DM are at a higher risk for incident NTM disease. The key findings were that (i) T2DM alone is not a statistically significant risk factor for NTM disease, but (ii) however, T2DM significantly increases the hazard of NTM disease development in the case of ≥2 diabetes-related complications.

In tuberculosis, several studies have elucidated that diabetes and diabetes control status affect several aspects, including clinical manifestation, treatment outcome, and recurrence as well as the development of tuberculosis (15, 16, 23, 24). In contrast, few studies have investigated the impact of diabetes on the occurrence or outcome of NTM disease except one recent study which showed that patients with DM are at higher risk for NTM-caused disease (21). In that study, Wang et al. reported that the incidence of NTM disease in patients with type 1 and T2DM was 1.43-fold greater than that in patients without DM after analyzing data from 136,736 patients with DM and from matched cases from the National Health Insurance program in Taiwan between 2000 and 2015 (21). Although Wang et al. found that the increased risk of incident NTM-caused diseases was observed in both patients with type 1 DM and those with T2DM, the risk of NTM disease development was not evident in our cohort composed of only T2DM patients. Though the reason for these disparate results is unclear, the severity of DM could be a possible explanation; the patients of the present study cohort were diagnosed with T2DM based on the ICD-10 (International Classification of Diseases, 10th Revision) code alone, irrespective of whether the patients received oral antidiabetes agents or insulin treatment.

There could be several plausible biologic explanations of the association of diabetes and risk for NTM disease. Pavlou et al. reported that long-term hyperglycemia impairs phagocytosis and bactericidal activity of macrophages, which is manifested as a reduced expression of interleukin-12 and diminished nitric oxide production (25). In addition, patients with DM have strongly reduced gamma interferon production (13), which is an essential cytokine for the eradication of the growth of NTM species (26). Hyperglycemia increases vulnerability to infection by degrading neutrophil and T-cell function, decreasing humoral immunity, and downregulating the secretion of inflammatory cytokines (27–29). Moreover, diabetic mice showed a diminished T-helper 1 adaptive immunity by lower expression of interferon and interleukin-12 and subsequent lower T-cell response to the *Mycobacterium tuberculosis* antigen (30). All of these findings strongly imply that DM directly suppresses the host innate and adaptive immune responses to NTM infection and increase susceptibility to NTM disease.

The present study assessed whether the development of NTM disease increased in accordance with the severity of diabetes by using DM-related complications as a proxy marker for DM severity similar to that in a previous study (16). Notably, we found that the risk of NTM disease was evident only in patients with T2DM who had more diabetes-related complications. A number of reports have shown that immunity is further impaired when diabetes is uncontrolled. For instance, it was found that diabetic patients who had uncontrolled glucose levels were more likely to have impaired T-cell dysfunction (31). In addition, Kumar et al. found that systemic levels of several proinflammatory cytokines showed a significant positive association with the glycated hemoglobin, which is a laboratory-evaluated parameter that directly reflects diabetes control (32). Moreover, bone marrow-derived macrophages generated under the conditions of long-term hyperglycemia showed several intrinsic changes, including impaired phagocytosis and bactericidal activity (25). As previous studies consistently reported that diabetes-related complications are significantly more prevalent in long-term uncontrolled diabetes than in controlled diabetes (33, 34), our results on the association between the number of diabetes-related complications and the occurrence of NTM disease appear to be convincing and are in line with the findings of an earlier study which reported that tuberculosis risk in patients with DM increased significantly with an increase in the number of diabetes-related complications (16).

Additionally, the duration of DM itself could be related to impaired immunity (28, 35). To analyze the impact of duration of T2DM on NTM disease development, we reanalyzed our data by dividing the matched pair into two groups: (i) 76,151 patients diagnosed with T2DM before 2007 and their matched pair and (ii) 115,067 patients with new-onset T2DM from 2008 to July 2019 and their matched pair (Fig. 1). The results of multivariable analyses in each group are shown in Tables S2 and 3 in the supplemental material and indicate that the relationship between the higher number of diabetes-related complications and the risk of NTM disease was evident only in those who were diagnosed with DM before 2007 (adjusted HR, 1.68; 95% CI, 1.18 to 2.38, $P = 0.004$ [Table S2]). These findings supported the hypothesis that patients with longer-duration T2DM were more likely to have an impaired immune system, which increased susceptibility to infectious disease, including NTM disease.

Approximately two-thirds of patients in the present study did not receive treatment for DM, and the rate seemed to be comparable to that of a previous study of an analysis of data from a National Health Insurance Service (NHIS) claim database from 2002 and 2013, which revealed that 53.9% of patients with T2DM did not use any antidiabetes drug (36). However, the proportion was lower than that in another study using data from the Korea National Health and Nutrition Examination Survey between 2016 and 2018, which reported that the proportion of DM patients receiving treatment with oral antidiabetes agents or insulin was approximately 60% (9). It was found that the proportion of patients receiving treatment in South Korea tended to gradually increase over time during 2007 to 2017 (37). Therefore, the low prescription rate in our study could be partially attributed to the fact that approximately 40% of our participants were diagnosed before 2007. In addition, it was possible that patients with IFG may have been misdiagnosed with DM, as a recent study in South Korea reported that the prevalence of IFG is nearly twice that of DM in the general population, which includes 27% of adults aged ≥30 years (9).

The present study had several limitations. First, we determined NTM disease on the basis of ICD-10 codes alone instead of microbiological data. This was because the detailed results of NTM culture data were not available in the NHIS-National Sample Cohort (NSC) data (38). Therefore, we were unable to confirm whether patients completely met the diagnostic criteria of NTM disease. In addition, the ICD-10 code of NTM disease alone cannot distinguish between pulmonary infection and other extrapulmonary disease. Therefore, we could not evaluate the burden of specific types of NTM disease in our study, although NTM pulmonary disease likely comprised the majority of the burden, as evinced in previous reports (39–41). Third, the ICD-10 codes for comorbidities other than NTM infection may have been unreported if the attending physician did not include the diagnostic codes in the insurance claim. Thus, it was possible that comorbidity may have been underestimated in our study participants. Moreover, in our cohort, the proportion of patients with comorbid diseases was higher in the T2DM group than in matched cohorts. Considering that patients with a higher number of comorbid diseases were likely to have visited hospitals more often than those who had fewer comorbidities, it was possible that these frequent visits increased the likelihood of diagnosis of NTM diseases. Although we adjusted all comorbid diseases in the multivariate analysis, there exists a possibility of further NTM detection in T2DM patients with a higher number of comorbidities. Finally, because the data concerning human immunodeficiency virus (HIV) are masked in the NHIS-NSC database, as this diagnostic code constitutes sensitive personal information (42), we were unable to adjust for HIV in our analysis. However, considering that the burden of HIV infection is low in South Korea (incident HIV infection rate in 2011 was 1.8 per 100,000 individuals [43]) and the prevalence of NTM disease was only approximately 2% (19/1,060) in HIV-infected patients in Southeast Asia (44), the cases of NTM disease associated with HIV infection would be low in our cohort.

In conclusion, this study involved an analysis of population-based cohort data in South Korea and showed an increased risk for NTM disease in T2DM patients with ≥2 diabetes-related complications. This finding suggested that T2DM patients with a larger number of complications should be regarded as a high-risk group for NTM disease.

## MATERIALS AND METHODS

**Data sources and ethics.** We used the National Health Insurance Service-National Sample Cohort 2.0 (NHIS-NSC) database, which is a population-based cohort that is maintained by the NHIS in South Korea (45). The NHIS is an obligatory national health care insurance system that covers 96.6% of the entire South Korean population (46). In the NHIS records, NSC comprises the anonymized data of 1,025,340 randomly selected participants, who comprised 2.2% of the total eligible South Korean population in 2002 (45). The NHIS-NSC was developed with an aim to provide representative information, such as medical history, including diagnosis codes, prescription details, and health screening results, to researchers and policymakers (45). The registered subjects were followed from 1 January 2002 to 31 December 2019.

From the NHIS-NSC, the data collected from 2007 to 2019 for the subjects were analyzed in the present study, as the reported incidence and prevalence of NTM disease were considerably low before 2007 in South Korea (47). We first excluded those (i) aged ≤19 years, (ii) with type I DM, (iii) who were already diagnosed with NTM before being diagnosed with DM, and (iv) who were diagnosed with NTM before 2007. After further exclusion of patients who were diagnosed with incident T2DM between July 2019 and December 2019 to ensure at least a 180-day observation period, 765,959 participants were eligible for inclusion in the analysis (Fig. 1), and the impact of T2DM on the subsequent development of NTM disease was assessed in these participants.

This study protocol was approved by the Institutional Review Board (IRB) of Asan Medical Center, Seoul, Republic of Korea (IRB no. 2020-3947). The requirement for informed consent was waived because the study used an existing database that was provided in a deidentified format.

**Matching of study participants.** Among the 765,959 adult participants aged ≥20 years in 2007, 76,151 (9.9%) patients were diagnosed with T2DM before 2007 (Fig. 1 and Table 1). These patients were first matched to those without DM in 2007 by using 1:1 matching for age and sex with a greedy caliper matching method (Fig. 1). Then, among the subjects without DM in 2007, we identified whether incident T2DM had developed during the follow-up period. Patients with new-onset T2DM were again matched with nondiabetic subjects in a 1:1 ratio according to age and sex, and this matching process was repeated for every year until 2018. In 2019, patients with T2DM were chosen and matched with nondiabetic subjects from January to June 2019. Using this matching process, we established the NTM-naive T2DM and the NTM-naive matched cohorts, and these patients were followed until the occurrence of NTM disease, death, or 31 December 2019, whichever came first.

**Definition of disease.** All diagnoses of the study participants were coded using the Korean Classification of Disease, Sixth Edition, which is a modified version of the International Classification of Diseases, 10th Revision (ICD-10).

We defined NTM disease as the presence of at least 1 claim of the diagnostic code associated with NTM infection (ICD-10 code A31x) (48, 49). The date of the first diagnosis code was defined as the index date of diagnosis of NTM disease. In addition, T2DM was determined by the presence of at least 1 claim of diabetes under ICD-10 codes E11 to E14 during the study period, regardless of whether the subject received treatment. The index date was determined as January 1 in 2007 for patients who were diagnosed with DM before 2007. For the remaining patients, the identified date of presence of the first claim for diabetes was defined as the index date.

Information on antidiabetes medication use was included and categorized into any oral antidiabetes agents and insulin. At least one claim for the prescription of antidiabetes medication within 1 year of claim of diabetes was assessed (50). The complications of DM included the following 6 categories (51): cardiovascular disease, nephropathy, retinopathy, peripheral vascular disease, stroke, and neuropathy, excluding metabolic disorder. In the present study, T2DM patients were considered to have complications if the ICD-10 code of a diabetes-related complication was identified within 1 year before and after the index date of T2DM. The severity of complications was analyzed by dividing the cohort into 2 categories according to the number of complications: ≤1 complication and ≥2 complications, as previously determined (16).

As described in the study enrollment process, we excluded patients with type 1 DM, who were defined as those who (i) had at least one claim under the ICD-10 code E10, without the code E11 before 2007, and (ii) received insulin without oral antidiabetes agents (50).

**Analysis of NTM disease development and matched cohort.** We analyzed whether the incidence of NTM disease differed over time according to the diagnosis and complications of T2DM during the follow-up period. We first compared the rates of incident NTM disease development between the NTM-naive T2DM cohort and the NTM-naive matched cohort. Next, we analyzed whether the occurrence of NTM disease differs according to the number of diabetes-related complication.

**Covariates.** We extracted data on a number of covariates from the NHIS-NCS, including age, sex, smoking status, various comorbidities (particularly bronchiectasis [38]), and the Charlson comorbidity index (CCI), which was quantified by using the ICD-10 code diagnoses for several major comorbidities, as previously defined (52). The presence of comorbidities and the measurement of CCI were assessed based on the time point within the 1 year preceding the index date of the diagnosis of T2DM.

**Statistical analysis.** Categorical variables were compared with McNemar's test (2 categories) and test of marginal homogeneity (3 categories), and continuous variables were compared with the Wilcoxon signed-rank test. Incidence rates of NTM disease per 100,000 person-years and 95% confidence intervals (CIs) were calculated in the NTM-naive T2DM and matched cohort. A Cox proportional hazard regression analysis was used to estimate the univariate and multivariable hazard ratios (HRs) for NTM disease among those in the NTM-naive T2DM cohort according to covariates, such as age and comorbidities. In addition, the differences of cumulative incidence of NTM according to T2DM and the complications were analyzed using the Kaplan-Meier method with a log rank test. Statistical analyses were performed using SAS EG

statistical software version 7.1 (SAS Institute, Cary, NC, USA) and R software version 3.3.3 (the R Foundation), and $P$ values of <0.05 were considered statistically significant.

**Data availability.** The data sets generated during and/or analyzed during the current study are not publicly available due to personal information protection.

## SUPPLEMENTAL MATERIAL

Supplemental material is available online only.

**SUPPLEMENTAL FILE 1**, PDF file, 0.1 MB.

## ACKNOWLEDGMENTS

No conflicts of interest are declared.

This work was supported by a National Research Foundation of Korea grant (no. 2022R1A2C1002847) that was funded by the Korea government (Ministry of Science and ICT) and by the Asan Institute for Life Sciences, Asan Medical Center, Seoul, Republic of Korea (grant no. 2021IT0002 to Kyung-Wook Jo).

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
