## [Reviewer comments · Microbiology Spectrum]

Microbiology Spectrum

Type 2 diabetes mellitus- and complications-related risk of nontuberculous mycobacterial disease in a South Korean cohort

Da Som Jeon, Seonok Kim, Mi-Ae Kim, Yong Pil Chong, Tae Sun Shim, Chang Hee Jung, Ye-Jee Kim, and Kyung-Wook Jo

Corresponding Author(s): Kyung-Wook Jo, Asan Medical Center, University of Ulsan College of Medicine

Review Timeline:

Submission Date:	November 6, 2022
Editorial Decision:	January 6, 2023
Revision Received:	February 28, 2023
Accepted:	March 3, 2023

Editor: Paschalis Vergidis

Reviewer(s): Disclosure of reviewer identity is with reference to reviewer comments included in decision letter(s). The following individuals involved in review of your submission have agreed to reveal their identity: Byung Woo Jhun (Reviewer #1)

Transaction Report:

DOI: <https://doi.org/10.1128/spectrum.04511-22>

January 6, 2023

Prof. Kyung-Wook Jo
Asan Medical Center, University of Ulsan College of Medicine
Division of Pulmonology and Critical Care Medicine, Department of Internal Medicine
Songpa-gu
Seoul
Korea (South), Republic of

Re: Spectrum04511-22 (Type 2 diabetes mellitus- and complications-related risk of nontuberculous mycobacterial disease in a South Korean cohort)

Dear Prof. Kyung-Wook Jo:

Link Not Available

Sincerely,

Paschalis Vergidis

Journals Department
Reviewer comments:

Reviewer #1 Please see attachment

Reviewer #2 (Comments for the Author):

The manuscript showed that type 2 DM with two or more complications increased the risk of NTM infection, using health insurance claim data. The topic is timely and the manuscript is well-written. However, there are some issues to be solved.

1. The authors used Cox model to measure the risk of NTM infection. As the authors stated in the discussion section, the reason

DM increases the risk of NTM infection might be explained in terms of immune insufficiency in patients with DM. As already known, this might be associated with the duration of DM. So, merging the patients with DM who were diagnosed before 2007 (who might have longer period of DM and might have more chance of complication) and the patients who were diagnosed with DM after 2007 seems to be unreasonable. What about the risk of NTM infection in patients with DM before 2007 and after 2007, respectively?

2. The authors adjusted the possible confounding factors. Then, what about HIV or other immune-suppressant use? As described in the manuscript, this showed the incidence of NTM infection (including disseminated NTM disease in HIV and NTM-PD, mainly in non-HIV hosts).

3. The CCI was much higher in patients with DM. So, even though adjusted in multivariate analysis, there might be the possibility of more chance of NTM-detection in patients with DM, as these had more chance of visiting hospitals.

4. How can you explain the exceptionally low rate of receiving treatment for DM? About two third of DM patients in South Korea do not receive DM medications according to the manuscript.

5. Metformin has been known to be used as a host-directed therapy in TB and has been involved in immune regulation in humans. What was the incidence of NTM in patients who received metformin vs other drugs?

Staff Comments:

Preparing Revision Guidelines

Please return the manuscript within 60 days; if you cannot complete the modification within this time period, please contact me. If you do not wish to modify the manuscript and prefer to submit it to another journal, please notify me of your decision immediately so that the manuscript may be formally withdrawn from consideration by Microbiology Spectrum.

Type 2 diabetes mellitus- and complications-related risk of nontuberculous mycobacterial disease in a South Korean cohort

This paper contains important information in NTM research area. But, there are a few things to point out.

Comment 1)

It would be better to describe the limitations of the study in more detail. Because, articles using diagnostic codes or big data have some unavoidable limitations. Because the authors did not use laboratory results, they cannot confirm whether patients met the ATS diagnostic criteria of NTM infection completely. In addition, NTM diagnostic codes cannot distinguish between pulmonary infection and other disease; so they cannot evaluate the burden of specific disease types. The diagnostic codes for comorbidities other than NTM infection may have been missed if the attending physician did not include the codes in the insurance claim. So, comorbidity may have been underestimated.

Comment 2)

The number of “NTM naïve T2DM cohort” and “NTM-naïve matched cohort” (n = 191,218) is exactly the same. But I think that, it is not an easy procedure to coordinate like this completely. Were there any errors or difficulties in this process?

Comment 3)

In this paper, there was a difference in the occurrence of NTM depending on the number of complications in DM patients. Could you explain in more detail from a pathophysiological point of view? Or do you think there is a relationship between the duration of DM and the occurrence of NTM?

February 28, 2023

Dr. Paschalis Vergidis

Editor

Microbiology Spectrum

Dear Dr. Paschalis Vergidis:

We thank the editors and reviewers for the comments on our manuscript titled “Type 2 diabetes mellitus- and complications-related risk of nontuberculous mycobacterial disease in a South Korean cohort” (manuscript ID: Spectrum04511-22). We have revised the manuscript in accordance with the suggestions and have provided our point-by-point responses to the reviewers’ comments below this letter. We believe that these changes have significantly improved the manuscript and hope that the manuscript is now acceptable for publication as a *Research Article* in *Microbiology Spectrum*.

Please note that we have changed the affiliation of Da Som Jeon (first author of the manuscript) from “Division of Pulmonary and Critical Care Medicine, Department of Internal Medicine, Asan Medical Center, University of Ulsan College of Medicine, Seoul, Republic of Korea” to “Division of Pulmonology and Critical Care Medicine, Department of Internal Medicine, Nowon Eulji Medical Center, University of Eulji, Seoul, Republic of Korea” to reflect the change in the author’s affiliation during the revision process.

Sincerely,

Ye-Jee Kim, PhD

Department of Clinical Epidemiology and Biostatistics, Asan Medical Center, University of Ulsan College of Medicine, 88, Olympic-ro 43-gil, Songpa-gu, Seoul 05505, Republic of Korea.

Email: kimyejee@amc.seoul.kr

AND

Kyung-Wook Jo, MD, PhD

Division of Pulmonary and Critical Care Medicine, University of Ulsan College of Medicine,
Asan Medical Center, 88 Olympic-ro 43-gil, Songpa-gu, Seoul 05505, South Korea

Phone: 82-2-3010-5783

Fax: 82-2-3010-6968

Email: siegliede@gmail.com

Responses to Reviewer #1:

This paper contains important information in NTM research area. But, there are a few things to point out.

→ Response: We sincerely thank you for identifying the areas of our manuscript that require revision and have accordingly made changes. We hope that the revised manuscript is now acceptable for publication.

Comment 1. It would be better to describe the limitations of the study in more detail. Because, articles using diagnostic codes or big data have some unavoidable limitations. Because the authors did not use laboratory results, they cannot confirm whether patients met the ATS diagnostic criteria of NTM infection completely. In addition, NTM diagnostic codes cannot distinguish between pulmonary infection and other disease; so they cannot evaluate the burden of specific disease types. The diagnostic codes for comorbidities other than NTM infection may have been missed if the attending physician did not include the codes in the insurance claim. So, comorbidity may have been underestimated.

Response 1.

Thank you for your valuable comments. We agree with your opinion.

1) In our study, we defined NTM disease as the presence of at least 1 claim of the diagnostic code associated with NTM infection (ICD-10 code A31x), as determined in previous studies conducted in South Korea (Kim et al. *Medicine* 2019;98:45, Yoon et al. *BMC Infectious Disease* 2017;17:432). Although another study (Lee et al. *Emerg Infect Dis* 2019;25:569-72) involving big data analysis in South Korea defined NTM infection as the presence of ICD-10 code A31 and claims data for acid-fast bacilli smears or mycobacterial culture, we did not include claims data for acid-fast bacilli smears or mycobacterial culture in our definition because detailed data on NTM culture results were unavailable in NHIS-NSC dataset.

This constraint has been described as a limitation of the present study in the Discussion

section of the original manuscript as follows;

“First, we determined NTM disease on the basis of ICD-10 codes alone instead of microbiological data. This was because the detailed results of NTM culture data was not available in NHIS-NSC data (Chest 159:1807-1811.).”

(Page 9, line 214–216 in the original / revised manuscript)

As reviewer pointed out, the major weakness of not including laboratory results in the definition of NTM disease was that we were unable to confirm whether patients completely met the ATS diagnostic criteria of NTM infection. Thank you for pointing this out. We had missed including this limitation in the original manuscript and have added the following sentence in the limitation section of the discussion:

“Therefore, we were unable to confirm whether patients completely met the diagnostic criteria of NTM disease.”

(Page 8, line 175–185 in the revised manuscript)

2) As reviewer correctly pointed out, we could not evaluate the burden of specific types of NTM disease in our study, because NTM diagnostic codes cannot distinguish between pulmonary infection and other diseases. However, we could cautiously assume that the majority of NTM disease in our study was likely to be NTM pulmonary disease as previous studies of NTM disease indicated that pulmonary NTM is the major component of the overall NTM disease burden. For instance, Ahn et al. recently reported that the proportion of pulmonary NTM disease comprised 89.6% of entire NTM disease incidence between 2007 and 2019 at a tertiary referral center in South Korea (Ahn et al. *Yonsei Med J* 2021;62:903-910). Another study in Canada showed that NTM from pulmonary sites comprised 96% of species/patient combinations isolated (Brode et al. *Emerg Infect Dis.* 2017;23:1898-1901). A study conducted in England, Wales, and Northern Ireland found that, among the culture-positive NTM isolates reported between 2007 and 2012, 90.9% were pulmonary isolates (Shah et al. *BMC Infectious Diseases* 2016;16:195).

To describe this point, we have added the following sentence in the Discussion section of the

revised manuscript:

“In addition, the ICD-10 code of NTM disease alone cannot distinguish between pulmonary infection and other extrapulmonary disease. Therefore, we could not evaluate the burden of specific types of NTM disease in our study, although NTM pulmonary disease likely comprised the majority of the burden, as evinced in previous reports (Chest 159:1807-1811.; Yonsei Med J 62:903-910.; Emerg Infect Dis 23:1898-1901).”

(Page 9, line 217–220 in the revised manuscript)

3) Concerning the comorbidities, we have added the following sentence to elucidate a limitation of the present study:

“Third, the ICD-10 codes for comorbidities other than NTM infection may have been unreported if the attending physician did not include the diagnostic codes in the insurance claim. Thus, it was possible that comorbidity may have been underestimated in our study participants.”

(Page 9, line 220–223 in the revised manuscript)

Comment 2. The number of “NTM naïve T2DM cohort” and “NTM-naïve matched cohort” (n = 191,218) is exactly the same. But I think that, it is not an easy procedure to coordinate like this completely. Were there any errors or difficulties in this process?

Response 2.

As Figure 1 indicates, the 191,218 sample consisted of two different matched cohorts during the study period; (i) the matched cohort of patients diagnosed with DM before 2007 (76,151 pairs), and (ii) the matched cohort of patients with new-onset T2DM diagnosed from 2008 to July 2019 (115,067 pairs).

First, concerning the matched cohorts of patients diagnosed with DM before 2007, the 76,151

patients with T2DM were matched to those without DM in 2007. As the number of patients without DM before 2007 was almost 10 times greater (n = 689,808, Figure 1), there was some statistical difficulty in the matching process.

In addition, from 2008 to July 2019, we generated 115,607 matched cohorts of patients with new-onset T2DM. Notably, the number of patients who were initially identified as having new-onset T2DM was 132,088 during this period (Figure 1). On matching patients with new-onset T2DM, 115,607 matched pairs were identified, after excluding 16,481 patients (Only 16,418 patients were excluded because we matched only for age and sex.). The matching process for these 115,607 pairs took up considerable time of the statisticians and required several attempts. However, we believe that we did not make any error in the matching process, as indicated by the distribution of age and sex between the two groups in Table 1.

Comment 3. In this paper, there was a difference in the occurrence of NTM depending on the number of complications in DM patients. Could you explain in more detail from a pathophysiological point of view? Or do you think there is a relationship between the duration of DM and the occurrence of NTM?

Response 3.

Thank you for your comments.

1) We have added the following paragraph to explain why the occurrence of NTM disease was evident in patients with diabetes who had a larger number of complications.

“Notably, we found that the risk of NTM disease was only evident in patients with T2DM who had more diabetes-related complications. A number of reports have shown that immunity is further impaired when diabetes is uncontrolled. For instance, it was found that diabetic patients who had uncontrolled glucose levels were more likely to have impaired T-cell dysfunction (Cell Metab 32:437-446.e5.). In addition, Kumar *et al.* found that systemic levels of several proinflammatory cytokines showed a significant

positive association with the glycated hemoglobin, which is a laboratory-evaluated parameter that directly reflects diabetes control (Ann Am Thorac Soc 10:441-449.). Moreover, bone marrow-derived macrophages generated under the conditions of long-term hyperglycemia showed several intrinsic changes, including impaired phagocytosis and bactericidal activity (BMC Immunol 19:24.). As previous studies consistently reported that diabetes-related complications are significantly more prevalent in long-term uncontrolled diabetes than in controlled diabetes (BMC Cardiovasc Disord 18:180.; Physiol Rev 93:137-188.), our results of the association between the number of diabetes-related complications and the occurrence of NTM disease appears to be convincing and is in line with the findings of an earlier study which reported that tuberculosis risk in patients with DM increased significantly with an increase in the number of diabetes-related complications (Clin Infect Dis 54:818-825.).”

(Page 8, line 178–191 in the revised manuscript)

2) In addition, in the Discussion section of the revised manuscript, we have added the following paragraph to describe the association of the duration of T2DM and the risk of NTM disease development as follows:

“Additionally, the duration of DM itself could be related to impaired immunity (Curr Diabetes Rev 16:442-449.; Diabetes Metab Syndr Obes 14:1897-1908.). To analyze the impact of duration of T2DM on NTM disease development, we re-analyzed our data by dividing the matched pair into two groups: (i) 76,151 patients diagnosed with T2DM before 2007 and their matched pair, and (ii) 115,067 patients with new-onset T2DM from 2008 to July 2019 and their matched pair (Figure 1). The results of multivariable analyses in each group are shown in Supplementary Tables 2 and 3, and indicate that the relationship between the higher number of diabetes-related complications and the risk of NTM disease was only evident in those who were diagnosed with DM before 2007 (adjusted HR 1.68, 95% CI 1.18–2.38, $P = 0.004$, Supplementary Table 2). These findings supported the hypothesis that patients with longer duration T2DM were more likely to have an impaired immune system, which increased susceptibility to infectious disease, including NTM disease.”

(Page 8–9, line 192–201 in the revised manuscript)

Responses to Reviewer #2:

The manuscript showed that type 2 DM with two or more complications increased the risk of NTM infection, using health insurance claim data. The topic is timely and the manuscript is well-written. However, there are some issues to be solved.

→ Response: We sincerely thank you for identifying the areas of our manuscript that require revision and have accordingly made changes. We hope that the revised manuscript is now acceptable for publication.

Comment 1. The authors used Cox model to measure the risk of NTM infection. As the authors stated in the discussion section, the reason DM increases the risk of NTM infection might be explained in terms of immune insufficiency in patients with DM. As already known, this might be associated with the duration of DM. So, merging the patients with DM who were diagnosed before 2007 (who might have longer period of DM and might have more chance of complication) and the patients who were diagnosed with DM after 2007 seems to be unreasonable. What about the risk of NTM infection in patients with DM before 2007 and after 2007, respectively?

Response 1.

Thank you for your valuable comments.

As the reviewer correctly pointed out, the patients diagnosed with T2DM before 2007 would evidently have a different duration of diabetes. Therefore, it could be reasonably assumed that patients whose T2DM was diagnosed before 2007 were at higher risk of NTM disease development than those diagnosed with new-onset T2DM between 2008 and 2019. We re-analyzed our data by dividing the matched pair into two groups; (i) 76,151 patients whose T2DM were diagnosed before 2007 and their matched pair, and (ii) 115,067 patients who had new-onset T2DM from 2008 to July 2019 and the matched pair (Figure 1). The results of

multivariable analyses in each group are as follows;

Table. Results of univariate and multivariable analyses of nontuberculous mycobacterial disease by type 2 diabetes mellitus and number of diabetes-related complications in the 76,151 patients diagnosed with diabetes before 2007.

Settings	Total numbers	Matched pair	NTM-naïve T2DM cohort			NTM-naïve matched cohort			Crude HR (95% CI)	P-value	Adjusted HR (95% CI)*	P-value
			NTM disease	Person-year	Incidence (95% CI)	NTM disease	Person-year	Incidence (95% CI)				
T2DM												
vs. no T2DM	152,302	76,151	344	874,848.7	39.32 (35.28–43.70)	272	891,701.2	30.50 (26.99–34.35)	1.29 (1.10–1.52)	0.002	1.07 (0.89–1.28)	0.463
T2DM with ≤1 complication												
vs. no T2DM	114,740	57,370	242	676,146.8	35.79 (31.42–40.60)	219	686,600.1	31.90 (27.81–36.41)	1.13 (0.94–1.35)	0.204	0.97 (0.72–1.19)	0.759
T2DM with ≥2 complications												
vs. no T2DM	37,562	18,781	102	198,701.9	51.33 (41.86–62.31)	53	205,101.1	25.84 (19.36–33.80)	2.00 (1.43–2.79)	<0.001	1.68 (1.18–2.38)	0.004

Table. Results of univariate and multivariable analyses of nontuberculous mycobacterial disease by type 2 diabetes mellitus and the number of diabetes-related complications in the 115,067 patients who were diagnosed with diabetes after 2007.

Settings	Total numbers	Matched pair	NTM-naïve T2DM cohort			NTM-naïve matched cohort			Crude HR (95% CI)	P-value	Adjusted HR (95% CI)*	P-value
			NTM disease	Person-year	Incidence (95% CI)	NTM disease	Person-year	Incidence (95% CI)				
T2DM												
vs. no T2DM	230,134	115,067	365	752,053.4	48.53 (43.68–53.78)	274	763,971.6	35.87 (31.74–40.37)	1.35 (1.16–1.58)	<0.001	1.14 (0.97–1.34)	0.113
T2DM with ≤1												
complication	162,886	81,443	232	538,873.2	43.05 (37.69–48.96)	177	545,353.1	32.46 (27.85–37.61)	1.33 (1.09–1.61)	0.005	1.19 (0.97–1.45)	0.093
vs. no T2DM												
T2DM with ≥2												
complications	67,248	33,624	133	213,180.2	62.39 (52.24–73.94)	97	218,618.5	44.37 (35.98–54.13)	1.41 (1.01–1.83)	0.011	1.09 (0.83–1.43)	0.521
vs. no T2DM												

As these Tables show, the association between the higher number of diabetes-related complications and the risk of NTM disease was only evident in those who were diagnosed with DM before 2007. These findings support the theory that patients who had longer duration T2DM were more likely to have an impaired immune system, which conferred increased susceptibility to infectious disease, including NTM disease.

In the Discussion section of the revised manuscript, we have added the following paragraph to describe the association of the duration of T2DM and the risk of NTM disease development:

“Additionally, the duration of DM itself could be related to impaired immunity (Curr Diabetes Rev 16:442-449.; Diabetes Metab Syndr Obes 14:1897-1908.). To analyze the impact of duration of T2DM on NTM disease development, we re-analyzed our data by dividing the matched pair into two groups: (i) 76,151 patients diagnosed with T2DM before 2007 and their matched pair, and (ii) 115,067 patients with new-onset T2DM from 2008 to July 2019 and their matched pair (Figure 1). The results of multivariable analyses in each group are shown in Supplementary Tables 2 and 3, and indicate that the relationship between the higher number of diabetes-related complications and the risk of NTM disease was only evident in those who were diagnosed with DM before 2007 (adjusted HR 1.68, 95% CI 1.18–2.38, $P = 0.004$, Supplementary Table 2). These findings supported the hypothesis that patients with longer duration T2DM were more likely to have an impaired immune system, which increased susceptibility to infectious disease, including NTM disease.”

(Page 8–9, line 192–201 in the revised manuscript)

In addition, we have added **Supplementary Tables 2 and 3**, which present the detailed crude and adjusted hazard ratios of the two groups.

Comment 2. The authors adjusted the possible confounding factors. Then, what about HIV or other immune-suppressant use? As described in the manuscript, this showed the incidence of NTM infection (including disseminated NTM disease in HIV and NTM-PD, mainly in

non-HIV hosts).

Response 2.

Thank you for your comments.

1) Concerning HIV, please note that the information concerning HIV is not available in the National Health Insurance Service-National Sample Cohort 2.0 (NHIS-NSC) database. This is because some diagnostic codes (e.g., HIV, mental and behavioral disorders due to psychoactive substance use, and schizophrenia, schizotypal and delusional disorders) were masked in the NHIS-NSC database because these diagnostic codes comprise sensitive personal information (BMC Cancer 2021;21:755).

Therefore, we could not present the data on HIV and failed to adjust HIV as a covariate in our study. This appears to be limitation of the present study. However, the burden of HIV infection is low in South Korea and the incident HIV infection rate in 2011 (the mid-point of the study period) was 1.8 per 100,000 individuals.

(https://www.unaids.org/sites/default/files/country/documents/ce_KR_Narrative_Report%5B1%5D.pdf) In addition, another study previously revealed that the prevalence of NTM disease was only approximately 2% (19/1,060) in HIV-infected patients in Southeast Asia (Am J Respir Crit Care Med 2012;185:981-988). Thus, it could be cautiously assumed that the cases of NTM disease associated with HIV infection would be low in our sample.

In the revised manuscript, we have added the following statements on the limitations of the present study.

“Finally, because the data concerning human immunodeficiency virus (HIV) is masked in the NHIS-NSC database, as this diagnostic code constitutes sensitive personal information (BMC Cancer 21:755.), we were unable to adjust for HIV in our analysis. However, considering that the burden of HIV infection is low in South Korea (incident HIV infection rate in 2011 was 1.8 per 100,000 individuals (https://www.unaids.org/sites/default/files/country/documents/ce_KR_Narrative_Report%5B1%5D.pdf. Accessed 2 February 2022.)) and the prevalence of NTM disease was

only approximately 2% (19/1,060) in HIV-infected patients in Southeast Asia (Am J Respir Crit Care Med 185:981-988.), the cases of NTM disease associated with HIV infection would be low in our cohort..”

(Page 10, line 228–234 in the revised manuscript)

2) When we extracted data from the raw data of NHIS-NCS, immunosuppressant use was not included among the selected variables. According to the rules of NHIS-NCS data use, only extracted data, but not raw data, are supposed to be archived after the data have been extracted. Therefore, information on immunosuppressant use is unavailable in our current extracted data set. To obtain data on immunosuppressant use, the investigators have to re-apply for approval from the NHIS-NCS by submitting an application form and re-submitting a research proposal. Therefore, the raw data, including data on immunosuppressant use, will be available again only if approval is granted. However, the number of researchers who can simultaneously access NHIS-NCS data is limited and, given the current situation, it will take several months (probably a year) after applying for NHIS-NCS access to extract the raw data. Accordingly, it is extremely difficult for us to receive the data again within the designated revision due date.

Therefore, we could not adjust for immunosuppressant use in our analysis. Considering that a previous study conducted in South Korea (Int J Endocrinol. 2020 Jul 24;2020:9879517) revealed that only approximately 2% of patients with type 2 DM had underlying autoimmune disease, which is the most common disease for prescription of immunosuppressants, we cautiously assumed that the number of patients treated with immunosuppressants were not substantially large enough to affect our main findings.

Comment 3. The CCI was much higher in patients with DM. So, even though adjusted in multivariate analysis, there might be the possibility of more chance of NTM-detection in patients with DM, as these had more chance of visiting hospitals.

Response 3.

→ Thank you for your valuable comments.

As suggested, the high rate of NTM development in our participants could be explained by the fact that patients with type 2 DM with a higher number of comorbid diseases visited the hospital often as compared with those who had fewer comorbidities. This could also explain why our study revealed a significant difference in a higher number of diabetes-related complications. Thank you for pointing this out.

We have accordingly added the following paragraph in the Discussion section of the manuscript.

“Moreover, in our cohort, the proportion of patients with comorbid diseases was higher in the T2DM group compared with matched cohorts. Considering that patients with a higher number of comorbid diseases were likely to have visited hospitals more often than those who had fewer comorbidities, it was possible that this frequent visit increased the likelihood of diagnosis of NTM diseases. Although we adjusted all comorbid diseases in the multivariate analysis, there exists a possibility of further NTM detection in T2DM patients with a higher number of comorbidities.”

(Page 9–10, line 223–228 in the revised manuscript)

Comment 4. How can you explain the exceptionally low rate of receiving treatment for DM? About two third of DM patients in South Korea do not receive DM medications according to the manuscript.

Response 4.

Thank you for your comments.

We have added the following paragraph to explain the low rate of treatment for DM in our study subjects.

“Approximately two-thirds of patients in the present study did not receive treatment for DM, and the rate seemed to be comparable to that of a previous study of an analysis of data from an NHIS claim database from 2002 and 2013, which revealed that 53.9% of patients with T2DM did not use any anti-diabetes drug (PLoS One 14:e0210159.). However, the proportion was lower than that in another study using data from the Korea National Health and Nutrition Examination Survey between 2016 and 2018, which reported that the proportion of DM patients receiving treatment with oral anti-diabetes agents or insulin was approximately 60% (Diabetes Metab J 45:1-10.). It was found that the proportion of patients receiving treatment in South Korea tended to gradually increase over time during 2007–2017 (Epidemiol Health 41:e2019029.). Therefore, the low prescription rate in our study could be partially attributed to the fact that approximately 40% of our participants were diagnosed before 2007. In addition, it was possible that patients with IFG may have been misdiagnosed with DM, as a recent study in South Korea reported that the prevalence of IFG is nearly twice that of DM in the general population, which includes 27% of adults aged ≥ 30 years (Diabetes Metab J 45:1-10.).”

(Page 9, line 202–213 in the revised manuscript)

Comment 5. Metformin has been known to be used as a host-directed therapy in TB and has been involved in immune regulation in humans. What was the incidence of NTM in patients who received metformin vs other drugs?

Response 5.

Thank you for this comment. Among DM patients, metformin use prevents tuberculosis and even decreases the tuberculosis-related mortality risk. This protective effect of metformin is mediated by an augmentation of the host’s innate immune response to *Mycobacterium tuberculosis* through various mechanisms, such as the promotion of autophagy. Despite a substantially increasing trend of NTM disease worldwide, including in South Korea, no study has investigated whether metformin use could decrease the risk for NTM disease in DM patients. We considered this topic another important issue concerning DM and NTM disease.

Therefore, we recently investigated whether patients with type 2 DM who were treated with metformin had a lower risk for NTM disease. That manuscript is “under review” with another journal, and the main method and results are summarized as follows:

From population-based retrospective cohort data of the Health Insurance Review and Assessment (HIRA) Service database of South Korea, which represents 97% of the entire South Korean population, we first extracted HIRA data of adult patients who were assigned an ICD-10 code for DM (ICD-10 E10–14) from January to July 2013. Next, we obtained the follow-up data of these patients until December 2016. The incident cases of NTM disease diagnosed based on the ICD-10 code (A31x) between January 2015 and December 2016 were included in the case group. Each case was matched to 4 controls (DM patients without incident NTM disease) according to the age, sex, Charlson Comorbidity Index, and insulin treatment. The preventive effect of metformin use on NTM disease was analyzed based on the difference in the metformin prescription rate between the case and control participants.

The proportions of patients who received metformin treatment were similar in the case and control groups (60.6% vs 55.7%, $p=0.116$). In addition, multivariate analysis showed no significant impact of metformin use on the risk of incident NTM disease. Therefore, we concluded that metformin use did not confer an NTM disease-preventive effect in DM patients.

Given the findings of that study, we inferred that prescription of metformin would not affect the main results of our manuscript. Please note, we could present the data only briefly because the other manuscript has not been published yet.

March 3, 2023

Prof. Kyung-Wook Jo
Asan Medical Center, University of Ulsan College of Medicine
Division of Pulmonology and Critical Care Medicine, Department of Internal Medicine
Songpa-gu
Seoul
Korea (South), Republic of

Re: Spectrum04511-22R1 (Type 2 diabetes mellitus- and complications-related risk of nontuberculous mycobacterial disease in a South Korean cohort)

Dear Prof. Kyung-Wook Jo:

Thank you for revising the manuscript and addressing the comments of the reviewers. I am impressed by your detailed response and your additional analysis that pertains to diagnosis of diabetes before and after 2007.

Your manuscript has been accepted, and I am forwarding it to the ASM Journals Department for publication. You will be notified when your proofs are ready to be viewed.

Sincerely,

Paschalis Vergidis
Editor, Microbiology Spectrum
